# Prediction of Change Rate of Settlement for Piled Raft Due to Adjacent Tunneling Using Machine Learning

**Dong-Wook Oh** [1], **Suk-Min Kong** [2], **Yong-Joo Lee** [1] **and Heon-Joon Park** [1,*]

1    Department of Civil Engineering, Seoul National University of Science and Technology, Seoul 01811, Korea; loeoh@seoultech.ac.kr (D.-W.O.); ucesyjl@seoultech.ac.kr (Y.-J.L.)

2    Future Infrastructure Research Center, Korea Institute of Civil Engineering and Building Technology, Goyang-si 10223, Gyeonggi-do, Korea; kongsukmin@kict.re.kr

\*    Correspondence: heonjoon@seoultech.ac.kr; Tel.: +82-2-970-6504

**Abstract:** For tunneling in urban areas, understanding the interaction and behavior of tunnels and the foundation of adjacent structures is very important, and various studies have been conducted. Superstructures in urban areas are designed and constructed with piled rafts, which are more effective than the conventional piled foundation. However, the settlement of a piled raft induced by tunneling mostly focuses on raft settlement. In this study, therefore, raft and pile settlements were obtained through 3D numerical analysis, and the change rate of settlement along the pile length was calculated by linear assumption. Machine learning was utilized to develop prediction models for raft and pile settlement and change rate of settlement along the pile length due to tunneling. In addition, raft settlement in the laboratory model test was used for the verification of the prediction model of raft settlement, derived through machine learning. As a result, the change rate of settlement along the pile length was between 0.64 and −0.71. In addition, among features, horizontal offset pile tunnel had the greatest influence, and pile diameter and number had relatively little influence.

**Keywords:** tunneling; piled raft; numerical analysis; machine learning; settlement

## 1. Introduction

Tunneling inevitably initiates ground deformations. This affects adjacent structures' behaviors, with variables such as the relative density of the ground, groundwater level, structure type, and tunnel location. Tunneling in urban areas is essential to understand the interactive behavior of pile foundation and adjacent structures, and many studies have been carried out on this through closed form, numerical analysis [1–4], 1 g laboratory model testing [5], centrifuge model testing [6,7], and field monitoring [8–10]. Among foundation types of structure, the piled raft is more rational than the conventional pile foundation, which does not consider the supporting load of the raft. In a piled raft, the pile is considered to act as a settlement reducer of the raft, and settlement in a piled-raft system is a very important factor in understanding the system.

When tunneling is not considered, since both the raft and the bottom of the pile show compression behavior, the difference between the settlement of the raft and the settlement of the pile may not be significant. If tunneling is considered, the amount of ground settlement varies depending on the distance from the tunnel, and it is also unclear whether ground deformation by tunneling acts in the same way as the raft does.

However, existing studies related to piled rafts were conducted without dividing the settlement into raft and pile; furthermore, only few variables were considered for the piled raft-tunneling problem. When the difference of settlement between the raft and pile due to tunneling is huge, the skin of the pile cracks, and has direct effect on the stability of the pile and structures. Thus, the aim of this study is important to understand the relationship between the difference of settlement between raft and pile tip and its impact on pile concrete.

Machine learning could be utilized to solve the problems mentioned earlier. Particularly as the geotechnical engineering field faces huge uncertainty concerning materials such as sand, clay, rock, etc., many researchers and engineers have tried to reduce the amount of difference or estrangement between numerical analysis, laboratory model tests and real phenomena on the construction site. For this matter, machine learning will be a very powerful and useful tool that could be utilized not only for uncertainty in geotechnical engineering, but also in medical [11,12], mechanical [13,14] and engineering fields, etc.

In this study, therefore, when tunneling adjacent to the piled raft was performed, a laboratory model test and numerical analysis were conducted to analyze settlement by separating raft and pile. In the laboratory model test, the settlement of the raft in the piled raft was measured, and in numerical analysis the settlement of raft and pile was measured. In addition, machine learning was utilized using settlement data derived from numerical analysis for the settlement prediction model. Through this model, the laboratory model test result was predicted and compared to verify the accuracy of the prediction model; on this basis, pile settlement for a piled-raft prediction model was derived. A model for predicting the rate of change of settlement according to the pile length was also developed.

## 2. Literature Review

### 2.1. Piled Raft

The load sharing ratio of the piled raft, unlike the existing pile foundation, which does not consider the bearing capacity of the raft, is a concept that separately considers the bearing capacity of the pile and the raft, as illustrated in Equation (1):

$$\alpha_P = \frac{Q_P}{Q_{PR}} = \frac{Q_P}{Q_P + Q_R} = 1 - \frac{Q_R}{Q_P}, \tag{1}$$

where $Q_P$ is pile bearing capacity, $Q_R$ is raft bearing capacity, and $Q_{PR}$ is piled raft bearing capacity.

The authors of [15] derived the normalized load-sharing ratio through centrifuge testing and finite-element analysis, and proposed interaction factor $\beta$. The load-sharing ratio can be obtained as shown in Equation (2), where $\eta_r = Q_r/Q_{ur}, \eta_p = Q_p/Q_{gp}$ represent the load-bearing coefficients of the raft and pile, respectively.

$$\alpha_P = \frac{1}{(\beta \times \xi) \left[ \frac{a_P \times \lambda_B + b_P \times (s/B_r)}{a_r + b_r \times (s/B_r)} \right] + 1} \tag{2}$$

Here, $s/B_r$ is foundation-relative settlement, and settlement is normalized to raft width ($B_r$); $a_r$, $b_r$, $a_p$, and $b_p$ are normalized model parameters, and 0.02, 0.8, 0.01, and 0.9 can be applied, respectively. $\xi$ is the same as in Equation (3), where $Q_{ur,u}$ is the unpiled raft, and $Q_{gp,u}$ is the ultimate load of the grouped pile.

$$\xi = \frac{Q_{ru,u}}{Q_{gp,u}} \tag{3}$$

In the present study, types of foundation, including single pile, unpiled raft, grouped pile, and piled raft, raft size, number, and spacing, and ground conditions were considered to be variables. The load-sharing ratio rapidly decreased to the point where the relative settlement was 0.02, and then converged. As the interaction factor increased, the load-sharing ratio value generally decreased. The relationship between relative settlement and interaction factor is shown in Equation (4).

$$\beta = 0.09(s/B_r)^{-0.32} \tag{4}$$

The authors of [16] analyzed pile axial load and bending moment for a disconnected piled raft using a centrifuge model test, and compared them with those of a connected

piled raft. According to this study, the disconnected piled raft significantly reduced the differential settlement and vertical settlement compared to the raft, which means that the disconnected piled raft is expected to be a better settlement reducer more than a connected piled raft. Here, settlement was measured at a shallow foundation because tunneling was not considered to be a variable for this study.

In [17], the authors developed a program to predict pile axial force, bending moment, and vertical and horizontal displacement due to tunneling adjacent to the pile foundation, and verified this through finite-difference analysis and closed form.

### 2.2. Machine Learning in Geotechnical Engineering Problems

Machine learning is an artificial-intelligence method that has recently been widely used for the prediction of regression and classification in academia and many industry sectors.

The authors of [18] modeled deep excavation through finite-element analysis and derived the prediction model of lateral deflection of the wall through machine learning. Extreme gradient boosting (XGB), support vector machine (SVM), multivariate adaptive regression spline (MARS), and artificial neural network (ANN) algorithms were used in this study. XGB was the best algorithm for the prediction of wall deflection. The RMSE of the XGB model was 5.518, where MARS, ANN, and SVM were 10.514, 11.281, and 16.614, respectively. RMSE is the most useful index for performance evaluation for regression problems in machine learning. Here, however, the verification of prediction models was not implemented through model test or filed monitoring.

In addition, [19,20] predicted ground-surface settlement was induced by tunneling using a machine-learning algorithm.

Research utilizing AI has limited variance due to its cost and the conditions of the research, and it can consider more variables than research utilized with regression analysis.

## 3. Laboratory Model Test

### 3.1. Apparatus for Laboratory Model Test

In this study, raft settlement for piled raft induced by tunneling was measured in the model test. Results were used for the verification of the prediction model, which was established through machine-learning algorithms. Prediction models were established only from numerical analysis; then, the model test result was compared with the value from the prediction model of raft settlement, which was obtained from numerical-analysis data only, for verification.

In this study, raft settlement due to tunneling under the piled raft was measured in the model test, and equipment, such as the model soil container and sand-pouring device, for the laboratory model test is shown in Figure 1. A sand-pouring device was used to form homogeneous ground for the model test, and by installing a soil-moisture content can, the relative density of the model soil was measured. As a result of the measured density of the model test, the relative density of the ground was about 34%–37%, which can be defined as loose sand according to the standards suggested by [21].

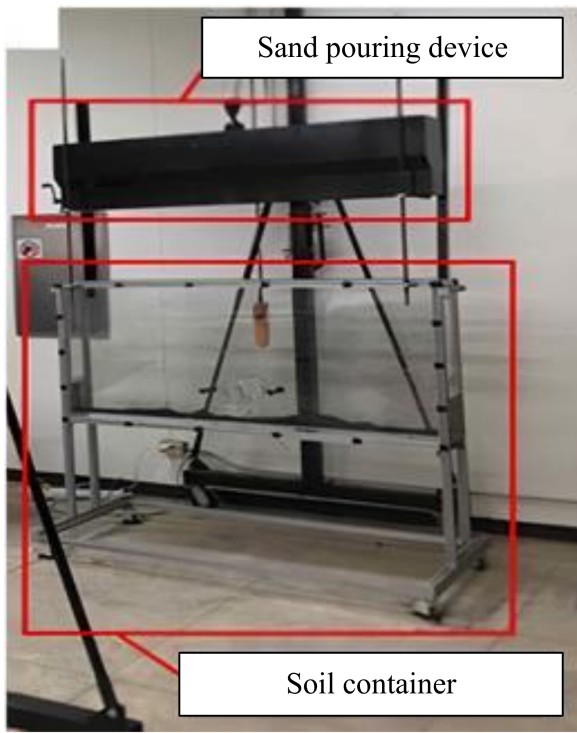

**Figure 1.** Devices for model test.

In the model test, 1/100 of the similarity law was applied to the model tunnel and the pile with length and diameter of 350 and 8 mm, respectively; the width and depth of the model raft were 70 and 20 mm, respectively. To simulate tunneling in the model test, a model tunnel was built, with a diameter of 100 mm. An overview of the model test is illustrated in Figure 2.

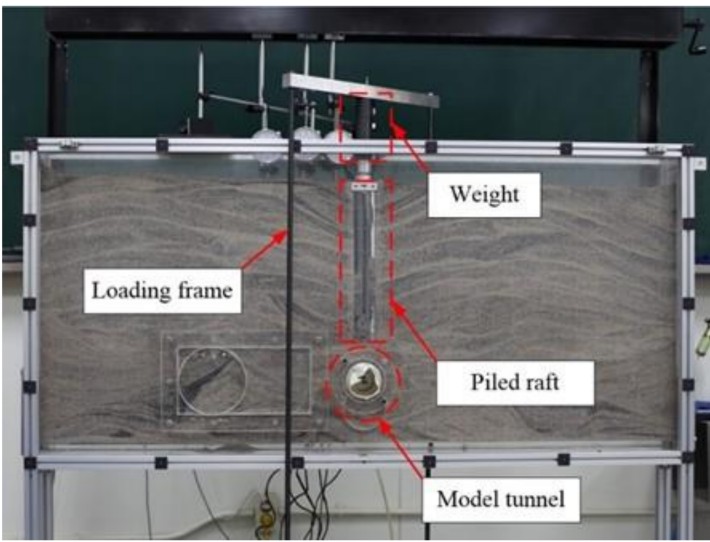

**Figure 2.** Overview of model test.

The allowed load implemented for this research, cited from [22].

### 3.2. Case Summary

The considered variables in the laboratory model test of this study were number of piles and vertical and horizontal pile toe–tunnel crown offsets, as summarized in Table 1. $P_N$ is the number of piles, and $O_V$ and $O_H$ are the vertical and horizontal offset of pile

toe–tunnel crown, respectively. On the basis of tunnel diameter (D), offsets were set to an $O_V$ of 0.5, 1.0, and 1.5 D, and $O_H$ of 0.0, 1.0, and 2.0 D.

**Table 1.** Summary of cases for laboratory model test.

| $P_N$ | Offset between Pile Toe and Tunnel Crown | | Relative Density | Note |
|---|---|---|---|---|
| | Horizontal | Vertical | | |
| 1 | 0.0 D | 0.5 D | L | 1_0.0H_0.5V_L |
| | | 1.0 D | L | 1_0.0H_1.0V_L |
| | | 1.5 D | L | 1_0.0H_1.5V_L |
| | 1.0 D | 0.5 D | L | 1_1.0H_0.5V_L |
| | 2.0 D | | L | 1_2.0H_0.5V_L |
| 2 | 0.0 D | 0.5 D | L | 2_0.0H_0.5V_L |
| | | 1.0 D | L | 2_0.0H_1.0V_L |
| | | 1.5 D | L | 2_0.0H_1.5V_L |
| | 1.0 D | 0.5 D | L | 2_1.0H_0.5V_L |
| | 2.0 D | | L | 2_2.0H_0.5V_L |
| 3 | 0.0 D | 0.5 D | L | 3_0.0H_0.5V_L |
| | | 1.0 D | L | 3_0.0H_1.0V_L |
| | | 1.5 D | L | 3_0.0H_1.5V_L |
| | 1.0 D | 0.5 D | L | 3_1.0H_0.5V_L |

### 3.3. Settlement of Piled Raft from Model Test

Piled-raft settlement due to tunneling in the laboratory model test is shown in Figure 3. Figure 3a shows the settlement of the piled raft as $O_H$ increased when $O_V$ = 0.5. When the $P_N$ was 1, settlement of 0.19 mm occurred at 0.0 D, and this decreased to 0.16 and 0.07 mm when the offsets increased to 1.0 and 2.0 D. When $P_N$ was 2, as $O_H$ increased from 0.0 to 1.0 and 2.0 D, settlement of the piled raft was 0.31, 0.12, and 0.1 mm, respectively. When $P_N$ was 3, there was settlement of 0.36 mm at $O_H$ of 0.0 D, 0.20 mm at 1.0 D, and 0.11 mm at 2.0 D.

Figure 3b shows the settlement of 1-, 2-, and 3-piled rafts as the $O_V$ increased from 0.5 to 1.0 and 1.5D when $O_H$ was 0.0 D. When $P_N$ was 1, piled-raft settlement of about 0.19 mm occurred at $O_V$ of 0.5 D, and it decreased to 0.15 and 0.14 mm as it increased to 1.0 and 1.5 D. When $P_N$ was 2, settlement of 0.31 mm occurred at 0.5 D, 0.19 mm at 1.0 D, and 0.18 mm at 1.5 D; settlement of 0.36, 0.30, and 0.27 mm occurred at 0.5, 1.0, and 1.5 D for the 3-piled raft, respectively.

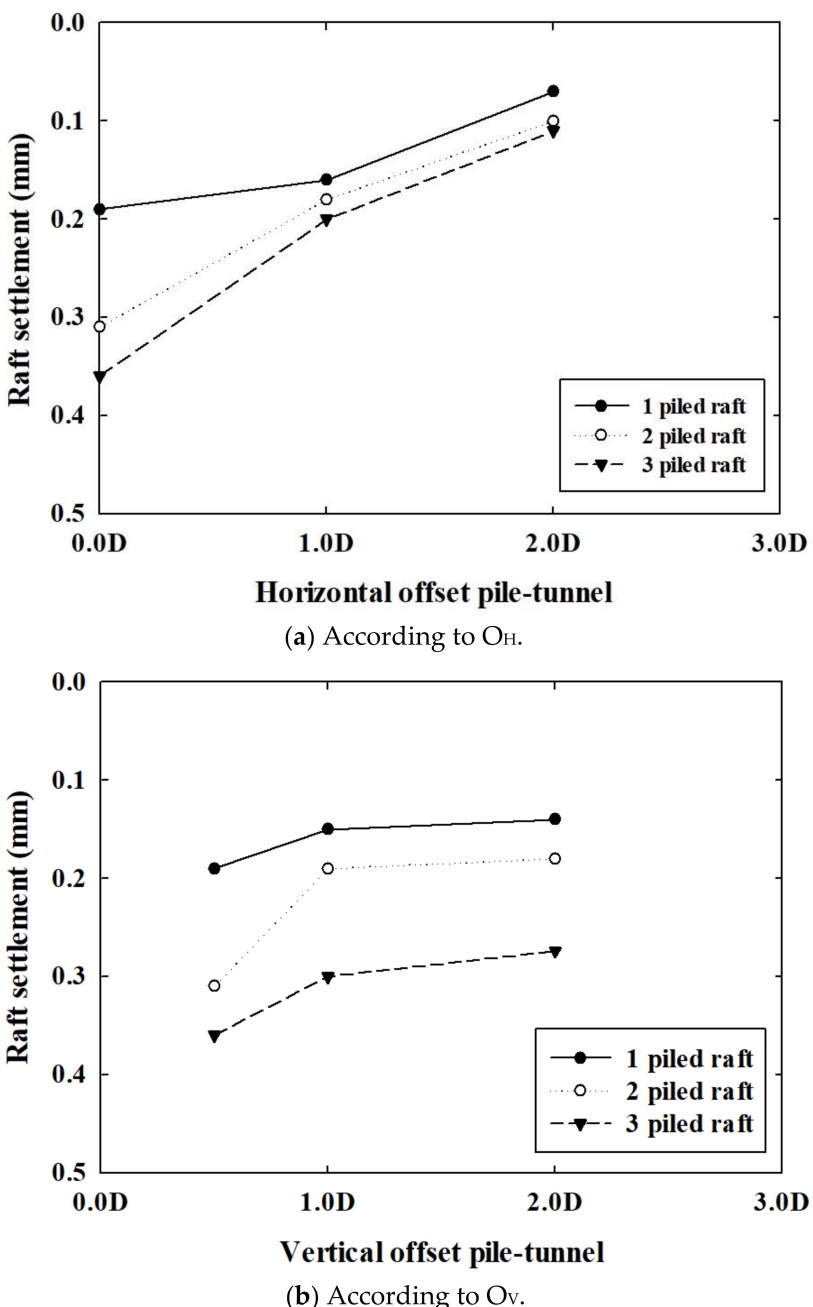

(**a**) According to O$_H$.

(**b**) According to O$_V$.

**Figure 3.** Settlement of piled raft due to tunneling from model test.

## 4. Finite-Element Analysis

### 4.1. Case Modelling and Summary

In this study, the raft and pile settlements of piled raft due to tunneling are analyzed, and numerical analysis was performed to derive a predictive model of the change rate of settlement along the length of pile.

Figure 4 shows the variables considered in this study and each settlement point for measurement. The settlement of raft ($S_R$) and pile ($S_P$) was measured in this numerical analysis; then, the change rate of settlement was calculated along pile length (dS). Change rate of settlement dS is defined as $(S_P - S_R)/P_L$, where $P_L$ is the length of the pile. As unit $S_P$ and $S_R$ were mm, the measurement unit for $P_L$ was m. It was also assumed that dS acted linearly from raft to pile toe; however, in reality, this could be a nonlinear relationship. This was further studied through additional model tests and numerical analysis.

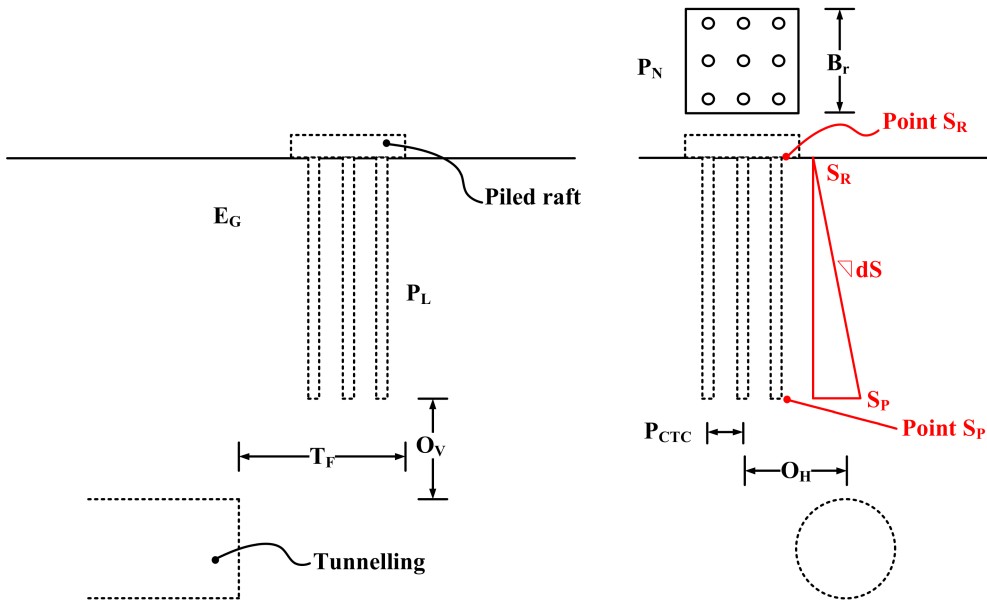

**Figure 4.** Problem considered for finite-element (FE) analysis.

The range and meaning of variables discussed in Figure 4 are summarized in Table 2, and mesh generation in FE analysis is shown in Figure 5. Tunneling is modeled using the concept of volume loss recommended by Atkinson

**Table 2.** Feature conditions for numerical analysis.

| Feature Name | Ranges | Comment |
|---|---|---|
| $E_G$ | 20e3, 40e3, 50e3 | Relative soil density |
| $P_N$ | 4, 9, 25 | Pile number |
| $O_V$ | 0.5, 1.5, 5.0 D | Vertical offset pile toe–tunnel crown (D = tunnel diameter) |
| $O_H$ | 0.0, 2.0, 5.0 D | Horizontal offset pile toe–tunnel crown (D = tunnel diameter) |
| $T_F$ | 10, 4, 3, 2, 1 Br | Horizontal distance of raft–tunnel face |
| $P_L$ | 6.5, 12.5, 22, 10, 20, 35, 12.5, 25, 44 | Pile length ($P_L/B_P$ = 13, 25, 44) |
| $P_{CTC}$ | 1.25, 2.5, 5.0 BP | Distance to center of piles |
| $B_P$ | 0.5, 0.8, 1.0 | Pile diameter |
| Br | 12, 20, 25 | Raft width |

Tunneling is modeled using the concept of volume loss that is mentioned by Atkinson [23]. The volume loss is a widely used method for the modeling of tunnel excavation in soil. In this study, the actual method of tunnel construction such as NATM, NTM or TBM is not applied because the aim of this study is to predict settlement of piled raft due to tunneling, as well as to verify this.

The allowable load applied to FE analysis was calculated by applying the safety factor after determining the ultimate load through the P–S curve of the unpiled raft and grouped pile. The ultimate load of the unpiled raft was calculated using the P–S curve, and the ultimate load was considered when settlement of about 25–30 mm occurred by referring to [24]. Grouped piles were calculated by classifying them according to pile diameter, using [25].

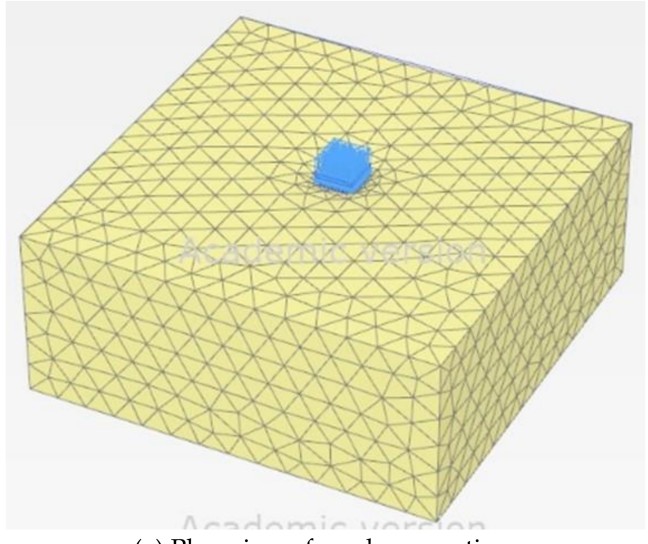

(**a**) Plan view of mesh generation.

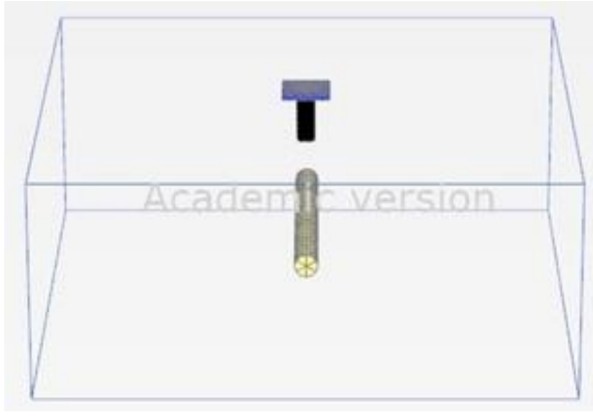

(**b**) Piled raft and tunnel.

**Figure 5.** Mesh generation of FE analysis.

### 4.2. Material Properties

In this study, the soil type and piled raft were assumed to be sandy soil and concrete, and the constitutive applied models were Mohr–Coulomb and linear elastic models. Although a more advanced constitutive model for ground material could be used, an additional laboratory model test for soil characteristics is needed to implement the advanced constitutive model. In this study, a laboratory model test for soil characteristics is not conducted and material properties are defined by previous literature. However, this will be considered as a variable for the further study with comparison to filed monitoring.

The relative density of the ground was divided into dense, medium and loose, and their unit weight ($\gamma$) and void ratio ($e$) are applied by Kim et al. [26]. Young's modulus ($E_G$), Poisson's ratio ($\nu$), cohesion ($c$), and shear resistance angle ($\phi$) were accordingly considered by Das [27]. Moreover, for the dilatancy angle for dense and medium shear resistance, angle $\phi$-30 was used, as proposed by [28]. The material properties considered in numerical analysis are summarized in Table 3.

**Table 3.** Material properties for FE analysis.

|  | Unit | Dense | Medium | Loose | Concrete |
|---|---|---|---|---|---|
| Constitutive model | - | MC | MC | MC | LE |
| Unit weight ($\gamma$) | kN/m$^3$ | 15.5 | 14.9 | 14.3 | |
| Void ratio ($e$) | - | 0.71 | 0.78 | 0.85 | - |
| Young's modulus ($E$) | kPa | 50e3 | 40e3 | 20e3 | 3.92e7 |
| Poisson's ratio ($\nu$) | - | 0.3 | 0.3 | 0.3 | 0.15 |
| Cohesion ($c$) | kPa | 0 | 0 | 0 | |
| Shear resistance angle ($\phi$) | degree | 40 | 35 | 30 | - |
| Dilatancy angle ($\psi$) | degree | 10 | 5 | 0 | - |
| $R_{inter}$ | - | 0.8 | 0.7 | 0.5 | - |

In Table 3, $R_{inter}$ is a factor that controls the strength of the structure and the surface of the ground; values were obtained from research by [29].

## 5. Machine Learning for the Prediction Model

### 5.1. Algorithm

Applications of artificial intelligence are rapidly increasing in various academic and industrial fields. Machine learning can be largely divided into supervised, unsupervised, and reinforcement learning, and this study was performed as supervised learning. The algorithms used in this study were extreme gradient boost (XGB) and multilayer perceptron (MLP).

XGB performs boosting and aggregating with a tree-based algorithm, such as decision tree and random forest, to extract training data and test data from input data to improve model performance. In addition, while sequentially repeating training prediction for several weak learners, if there is a difference between prediction value and input data, weight is given to calculate the error, which is faster than the traditional gradient boost method (GBM). The greatest advantage of XGB is that overfitting can be limited through its own function, and feature importance can be quantified. Feature importance refers to the importance of those features that influence the prediction model, and a tree-based algorithm can quantify this. XGB is the most powerful algorithm, and feature importance derived from it is the most reliable.

The MLP algorithm is representative based on artificial neural networks. Data were inputted to the perceptron of the input layer, and weight was assigned to the hidden layer. Therefore, weights applied in each hidden layer are unknown to the user. More detail about these algorithms can be found in [30,31].

### 5.2. Data Analysis and Preprocessing

Data analysis and preprocessing for input datasets are required to derive satisfactory precession models through machine learning. Prior to machine learning, the distribution, basic statistics, and null data checking and correction were identified among features of input data derived through numerical analysis.

Table 4 shows the basic statics of the input dataset, where $S_R$, $S_P$, and dS are raft settlement, pile settlement, and change rate of settlement along the pile length, respectively.

**Table 4.** Basic statics value of input data.

|  | Count | Mean | Std | Min. | 25% | 50% | 75% | Max. |
|---|---|---|---|---|---|---|---|---|
| $E_G$ | 2160 | 36,666.67 | 12,475.08 | 20,000 | 20,000 | 40,000 | 50,000 | 50,000 |
| $O_H$ | 2160 | 14.0 | 19.60 | 0.00 | 0.00 | 0.00 | 20.00 | 50.00 |
| $O_V$ | 2160 | 16.0 | 17.44 | 5.00 | 5.00 | 5.00 | 15.00 | 50.00 |
| $B_P$ | 2160 | 0.77 | 0.21 | 0.50 | 0.50 | 0.800 | 1.00 | 1.00 |
| $P_L$ | 2160 | 20.83 | 11.64 | 6.50 | 12.50 | 20.00 | 25.00 | 44.00 |
| $B_r$ | 2160 | 19.0 | 5.36 | 12.00 | 12.00 | 20.00 | 25.00 | 25.00 |
| $P_N$ | 2160 | 11.13 | 8.30 | 4.00 | 4.00 | 9.00 | 13.00 | 25.00 |
| $P_{CTC}$ | 2160 | 2.04 | 1.28 | 0.63 | 1.25 | 1.63 | 2.50 | 5.00 |
| $T_F$ | 2160 | 3.33 | 3.25 | 0.00 | 1.00 | 2.50 | 4.00 | 10.00 |
| $S_R$ | 2160 | 35.28 | 18.71 | 16.58 | 25.15 | 30.57 | 38.86 | 192.66 |
| $S_P$ | 2160 | 34.01 | 17.53 | 7.43 | 24.41 | 30.05 | 38.08 | 161.57 |
| dS | 2160 | −0.04 | 0.36 | −4.52 | −0.10 | −0.02 | 0.09 | 1.45 |

In Table 4, Count is the number of rows, Mean is the average of each feature, and Std is standard deviation. In addition, $E_G \sim T_F$ was used as an independent variable feature, and $S_R$, $S_P$, and dS were used as dependent variables.

Figure 6 visualizes the correlation between each feature and $S_R$, $S_P$, and dS.

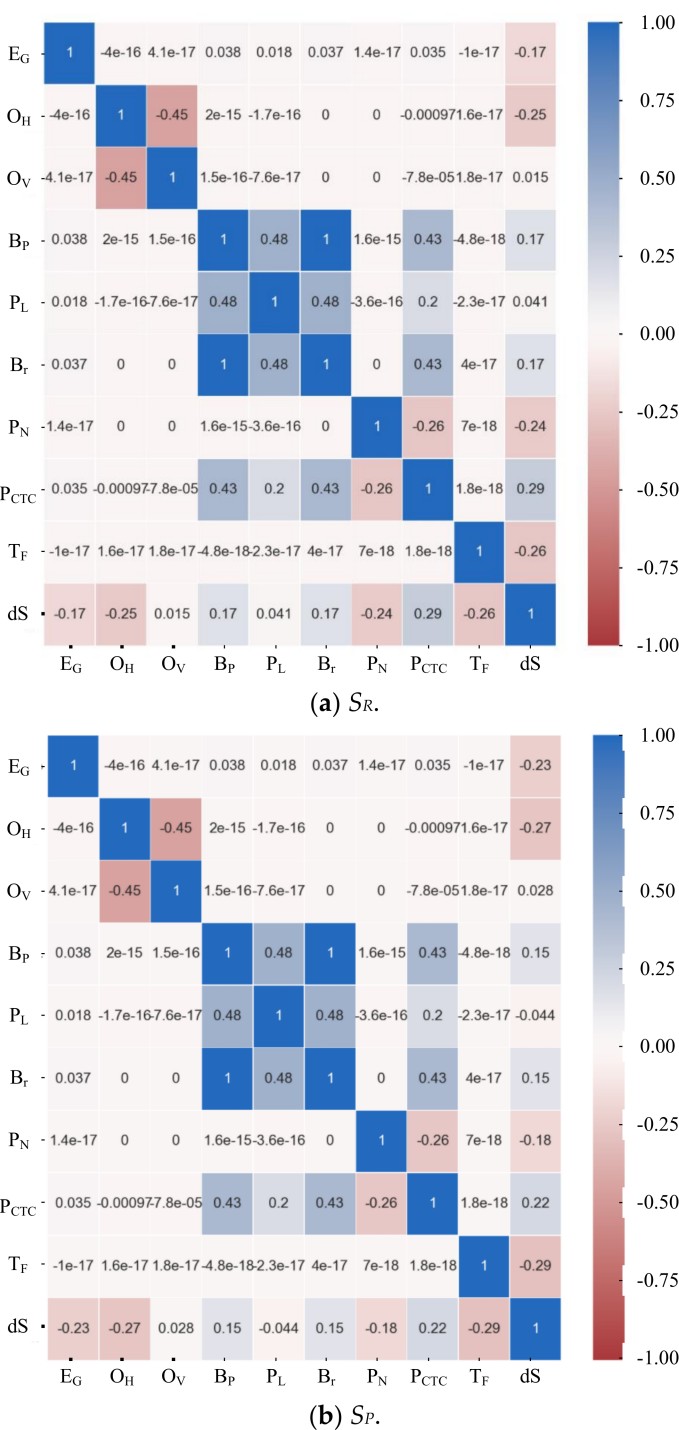

(**a**) $S_R$.

(**b**) $S_P$.

**Figure 6.** *Cont.*

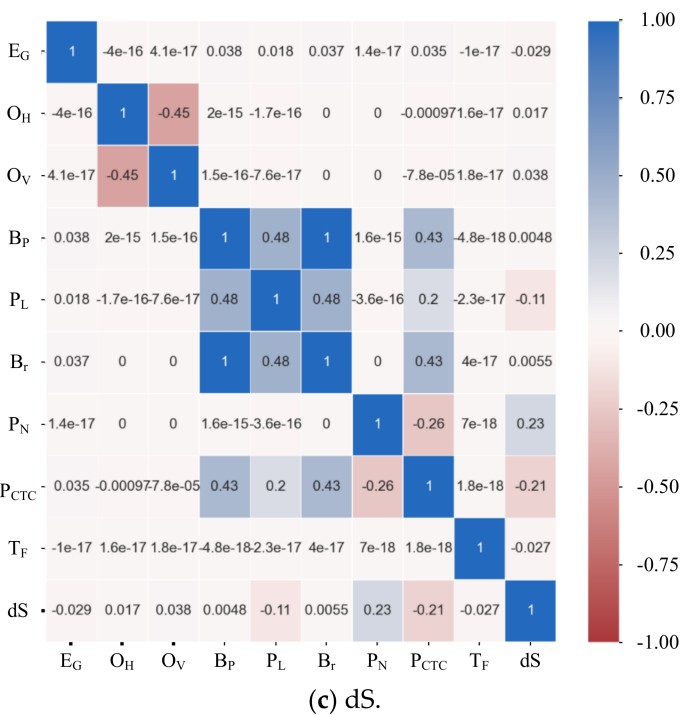

(**c**) dS.

**Figure 6.** Visualization of correlation of feature-dependent variables.

If correlation between two variables is positive, this indicates that variable 2 increases as variable 1 increases, and vice versa. In addition, being closer to 0 signifies the independence of each variable and being closer to 1 implies higher correlations.

Figure 7 shows the distribution of raw and regularization data in $S_R$, $S_P$ and dS, respectively. It is important to regularize aw data as normal distribution to improve performance of the prediction model.

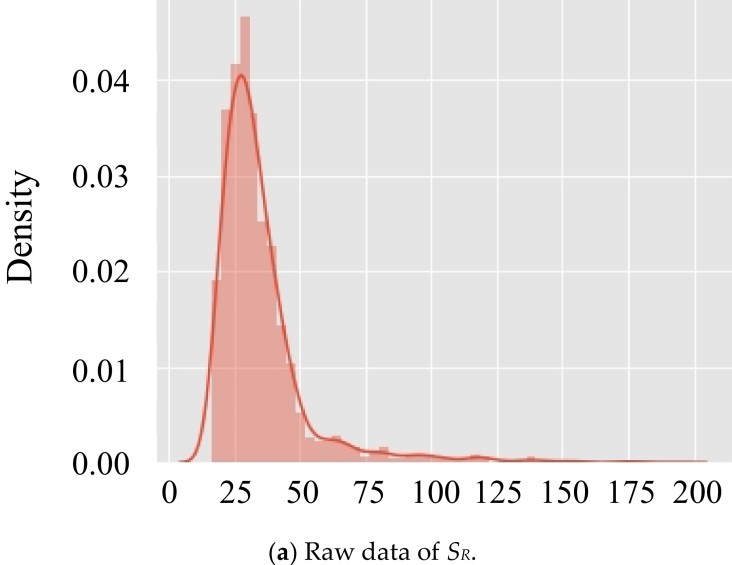

(**a**) Raw data of $S_R$.

**Figure 7.** *Cont.*

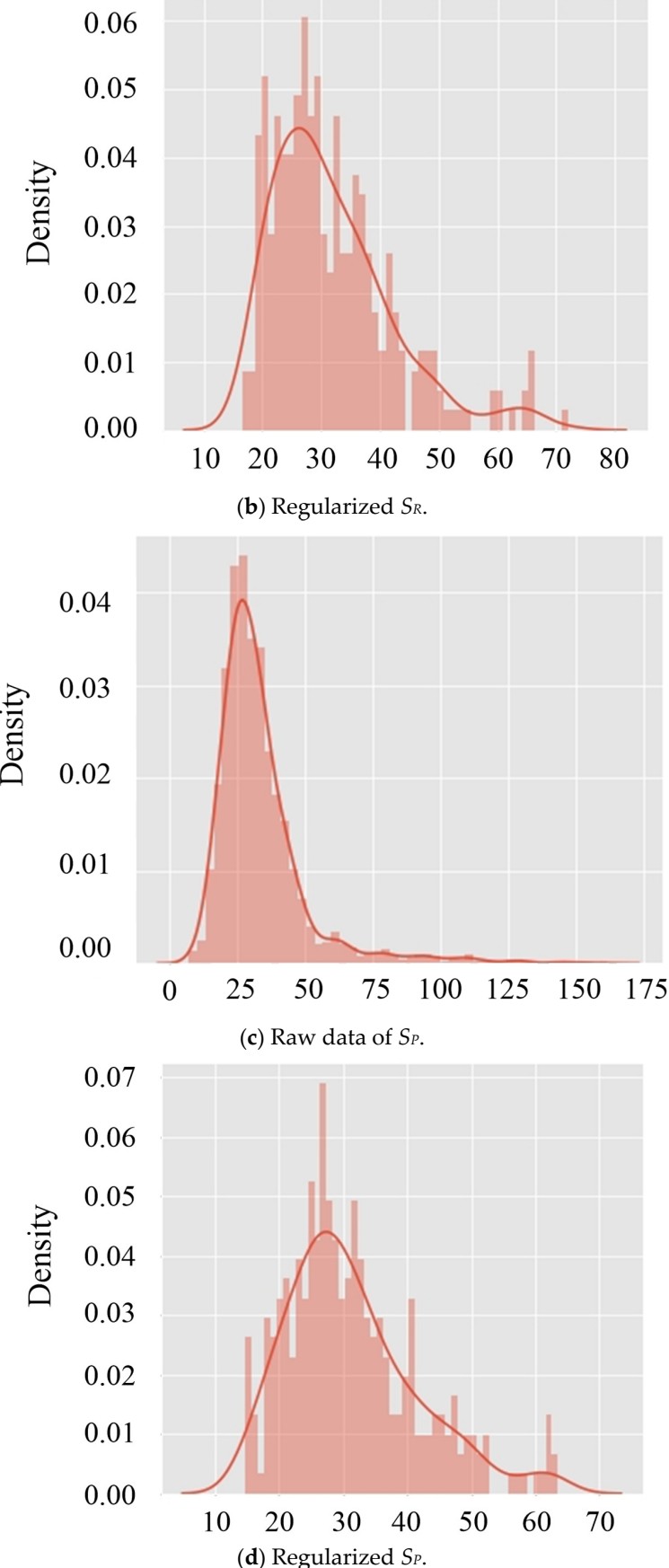

(**b**) Regularized $S_R$.

(**c**) Raw data of $S_P$.

(**d**) Regularized $S_P$.

**Figure 7.** *Cont*.

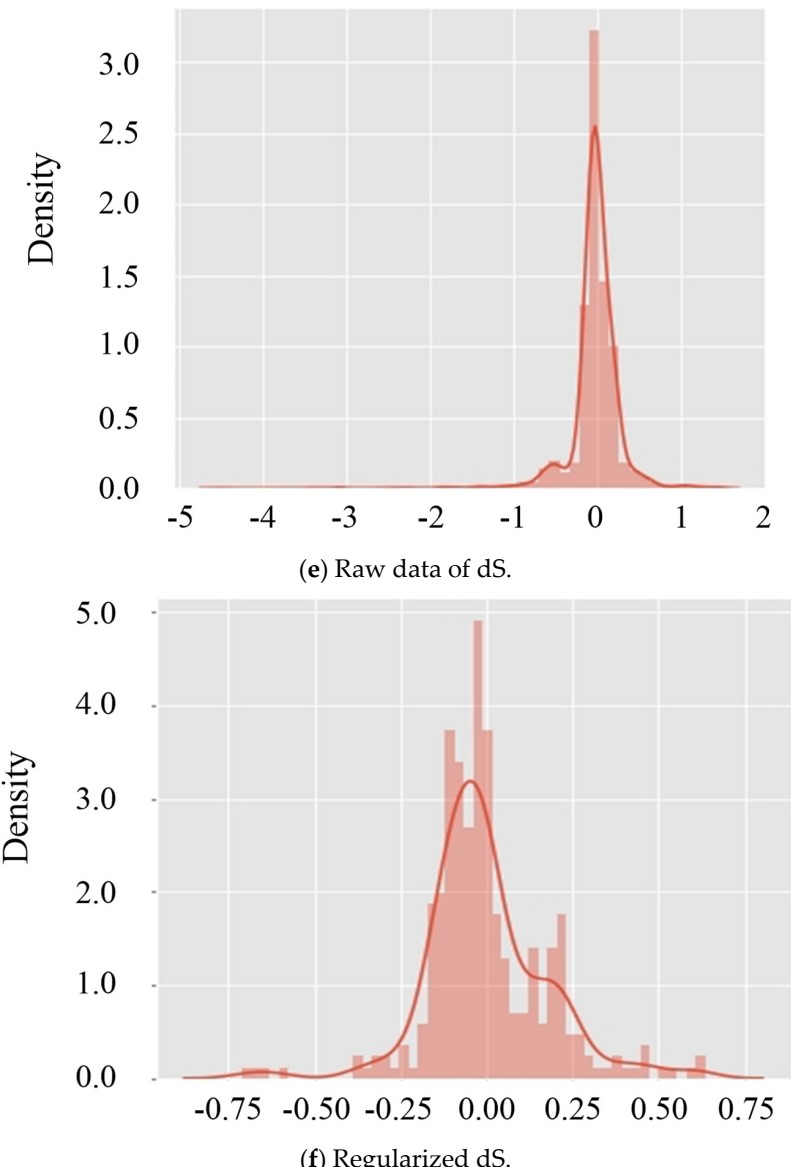

(**e**) Raw data of dS.

(**f**) Regularized dS.

**Figure 7.** Regularization of data dS.

*5.3. Results from Machine Learning*

5.3.1. Evaluation of Prediction-Model Performance

In the regression approach, machine learning and deep learning were utilized to generate the prediction model. The method for evaluating the performance of the prediction models is generally via mean absolute error (MAE), mean-square error (MSE), and root-mean-square error (RMSE) methods.

MAE is a concept that adds to the difference of the predicted and real values of the model, which is represented in Equation (5), and can intuitively represent the performance of the model. However, when comparing models with significantly different sizes of average error value, the prediction models favor those with smaller average errors.

$$MAE = \frac{1}{n}\sum|\hat{y} - y| \tag{5}$$

where, $\hat{y}$ is the predicted value, $y$ is the actual value, and $n$ stands for the number of input data.

MSE is a widely used index in the field of digital-image processing along with machine learning. MSE is calculated to obtain an average value by squaring the difference of the predicted and actual values, divided by numbers of data, as represented in Equation (6).

$$MSE = \frac{1}{n}\sum_{i=1}^{n}(\hat{y}_i - y_i)^2 \tag{6}$$

RMSE is the square root of MSE that is used to compare the error of the prediction model and determine the error interval of the prediction value, as indicated in Equation (7).

$$RMSE = \sqrt{MSE} = \sqrt{\left(\sum_{i=1}^{n}(\hat{y}_i - y_i)^2\right)/n} \tag{7}$$

MAE can represent the performance of the model by the summation of the difference value between the predicted and true values of the model. Upon evaluating the performance of a prediction model using mean error, the result is expressed as a negative number. If the prediction-model value is smaller than the true value, it is utilized to overcome the underestimation of the mean error values.

In this study, the evaluation of the prediction model was carried out by RMSE. Figure 8 represents the distribution of prediction models $S_R$ from XGB and MLP. When MLP was used compared to XGB, it was predicted to be larger in the 20 to 30 mm section, but smaller in the 50 to 60 mm section.

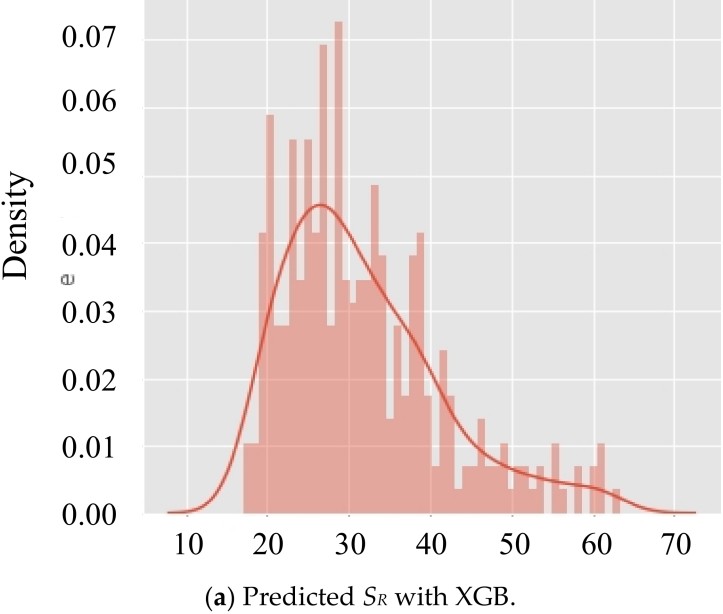

(**a**) Predicted $S_R$ with XGB.

**Figure 8.** *Cont.*

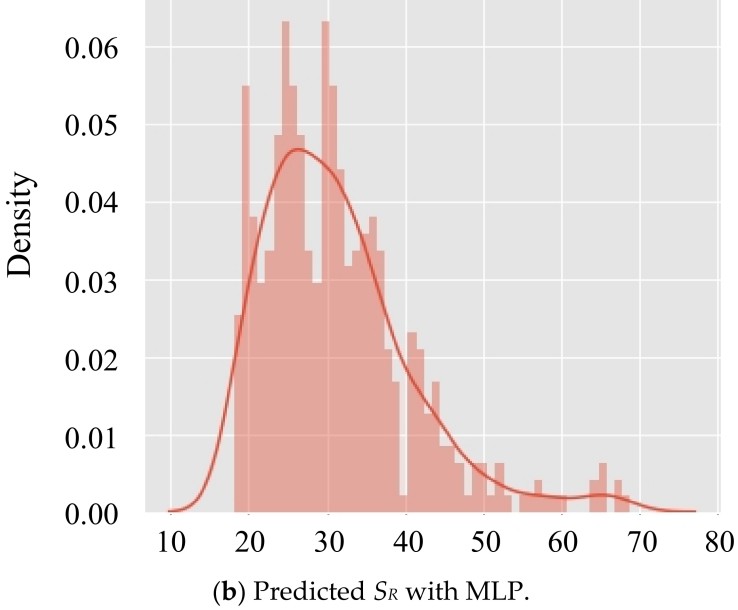

(**b**) Predicted $S_R$ with MLP.

**Figure 8.** Distribution of data and prediction model_$S_R$.

Each algorithm is schematically illustrated in Figure 9, so that it can be more intuitively analyzed.

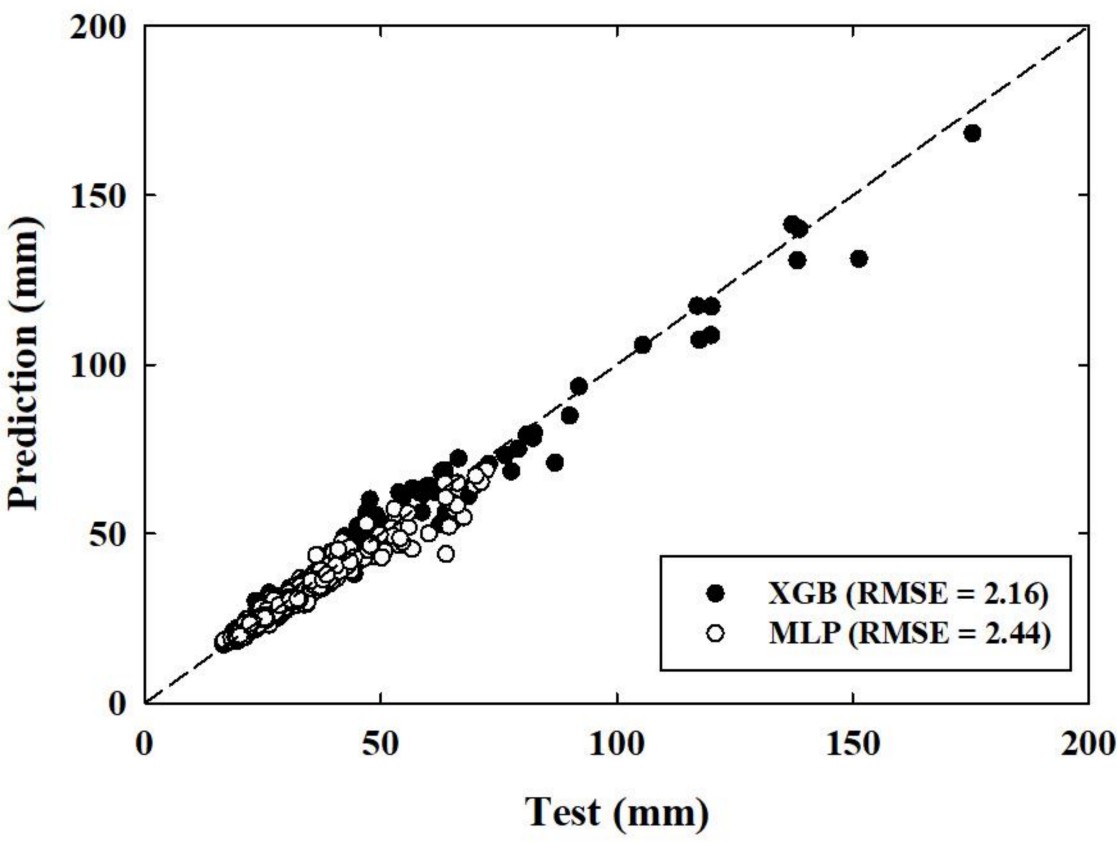

**Figure 9.** Prediction model_$S_R$.

If test data and prediction data had similar values, the dot was located on the dashed line, and the RMSE values of the prediction model_$S_R$ derived by XGB and MLP were 2.16 and 2.44, respectively.

Figure 10 shows the distribution of prediction data derived through XGB and MLP for prediction model_$S_P$. The section with the highest density (25–30 mm) appeared to be wider in MLP than that in XGB.

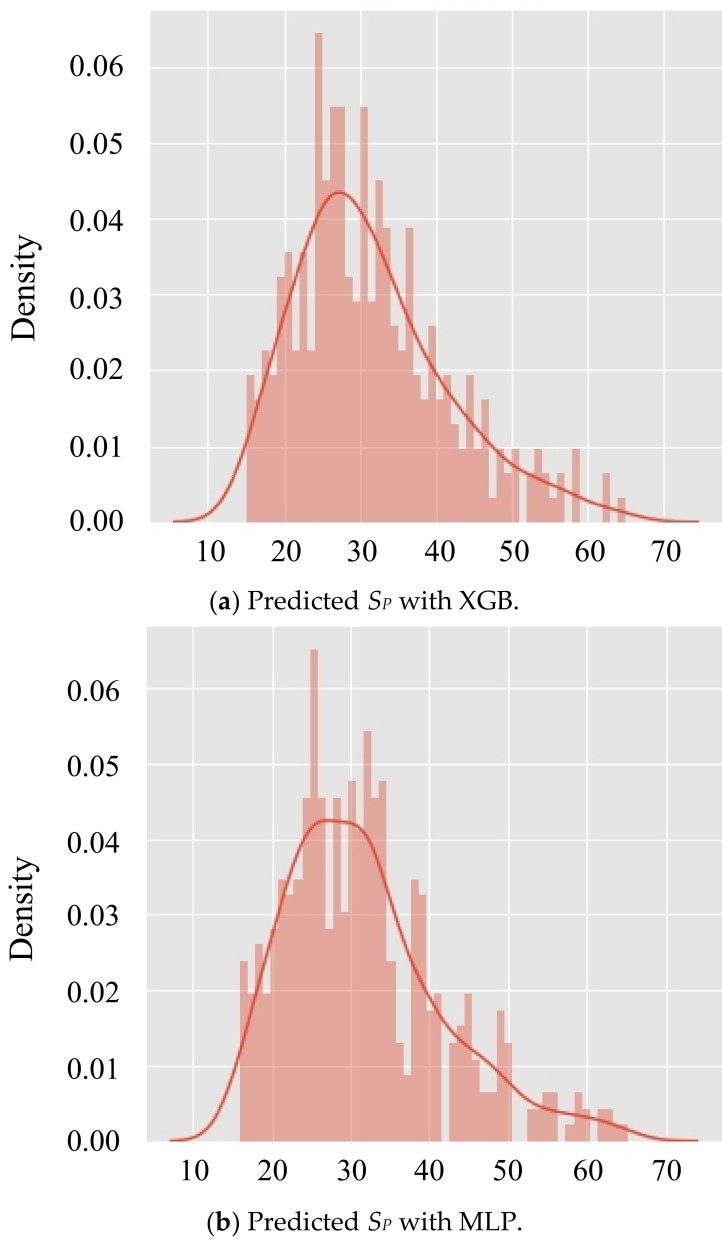

(**a**) Predicted $S_P$ with XGB.

(**b**) Predicted $S_P$ with MLP.

**Figure 10.** Distribution of data and prediction model_$S_P$.

Figure 11 compares the predicted data and test data of prediction model_$S_P$ derived from XGB and MLP. For the XGB algorithm, the RMSE value was about 1.63, and the MLP algorithm showed an RMSE value of 2.32. Similarly to prediction model_$S_R$, XGB showed slightly superior performance.

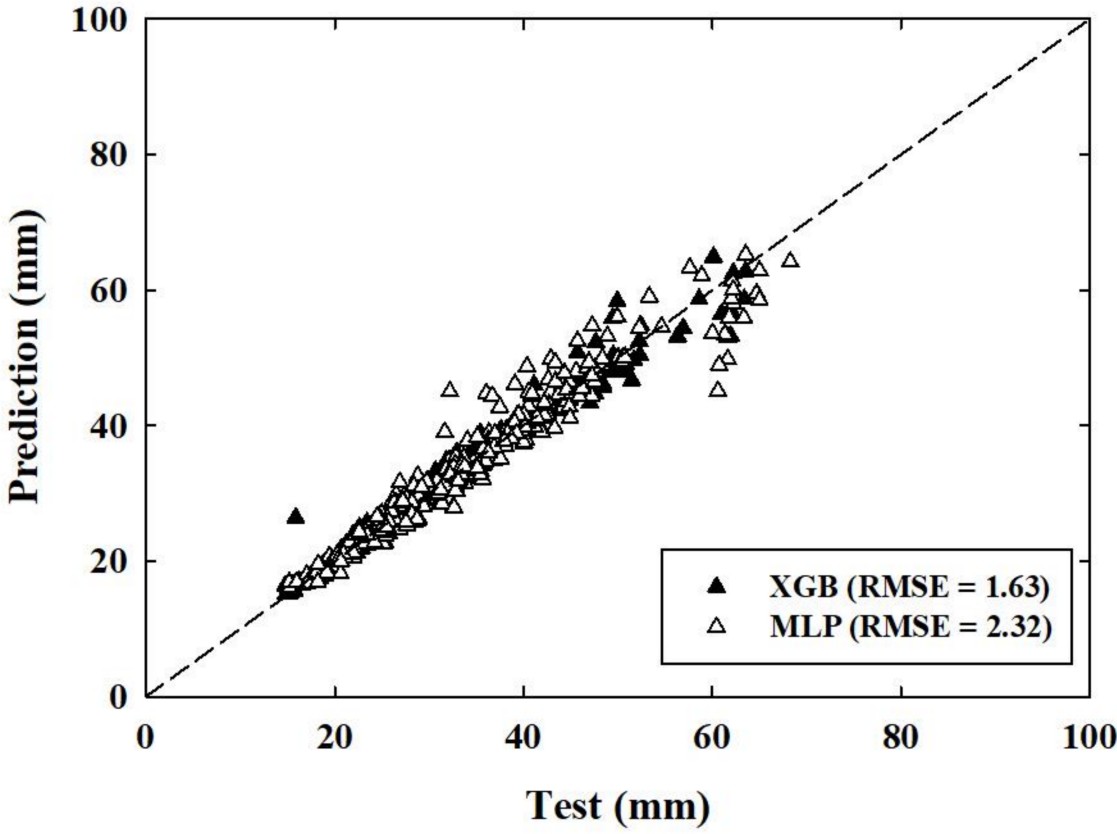

**Figure 11.** Prediction model_$S_P$.

Prediction model_dS was derived on the basis of the results of prediction mode_$S_R$ and prediction model_$S_P$; data distribution is shown in Figure 12. For XGB, dS values were distributed at the highest density values at 0.0–0.05 intervals. For MLP, the highest density was found at a smaller value.

Figure 13 shows a comparison of the prediction data and test data derived using XGB and MLP. The A zone is an area where the $S_P$ value is greater than $S_R$ due to tunnel excavation; the maximal value for XGB was 0.63, and for MLP was 0.64. The B zone is an area where $S_P$ is less than $S_R$ despite tunnel excavation, and the dS value is negative, with a minimal value of −0.71 to 0.60. With the XGB algorithm, the model's RMSE was 0.05, and the MLP showed an RMSE value of 0.07. As in the two previous models, prediction model_dS also showed better performance with XGB.

In prediction models using machine learning, overfitting and underfitting are problems that must be avoided.

Although overfitting is highly accurate, its usability is poor, and it requires much time to derive a predictive model. Underfitting has the problem of low model accuracy. Overfitting can be assessed by determining the RMSE values in the training data and test data, and if the RMSE in the training data is smaller than that in the test data, the model is considered to be overfitted.

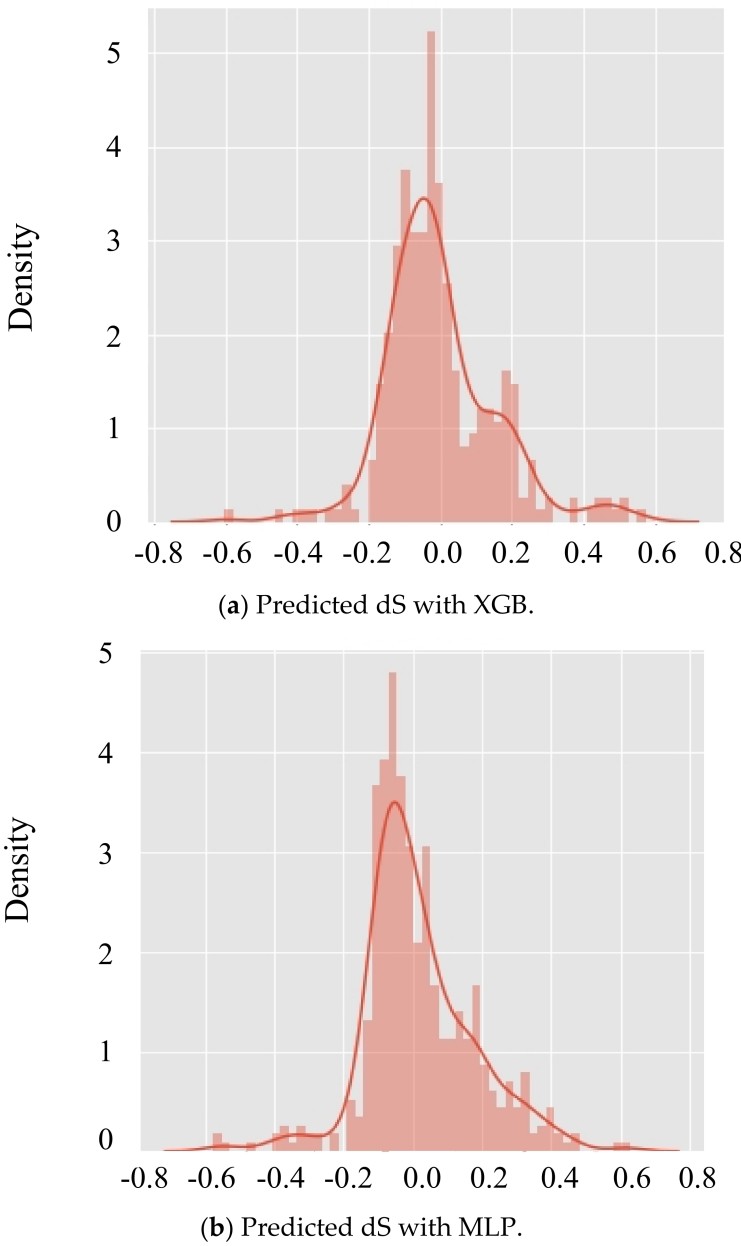

**Figure 12.** Distribution of data and prediction model_dS.

The underfitting case is determined by whether the RMSE of the derived model performances is acceptable. In Table 5, the training data of each algorithm and the RSME of the test data are compared. The RMSE of the training data was larger than that of the test data, so overfitting was avoided.

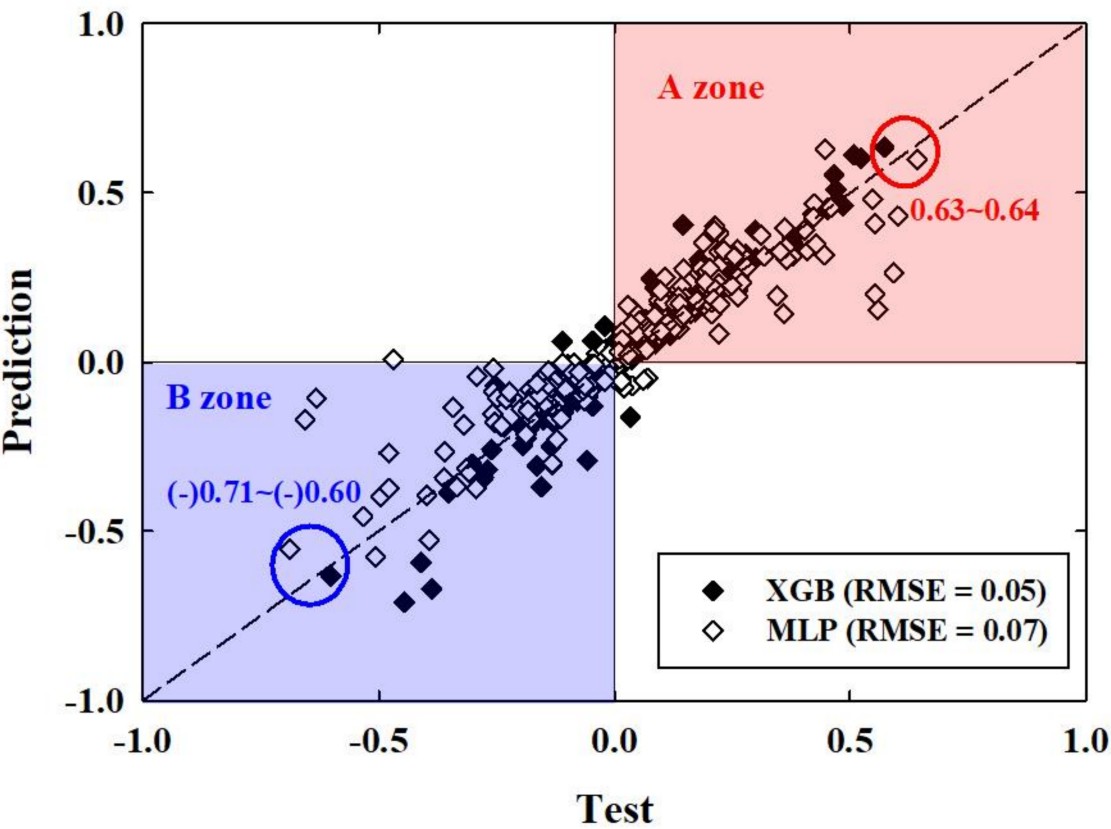

**Figure 13.** Prediction model_dS.

**Table 5.** Comparison of RMSE for prediction models.

| Prediction Model | XGB | | MLP | |
|---|---|---|---|---|
| | **Training** | **Test** | **Training** | **Test** |
| $S_R$ | 2.156 | 0.039 | 2.440 | 0.044 |
| $S_P$ | 1.632 | 0.030 | 2.321 | 0.043 |
| dS | 0.048 | 0.035 | 0.075 | 0.055 |

5.3.2. Verification of Prediction Model for Raft Settlement

In this study, raft settlement ($S_R$) and pile settlement ($S_P$) were predicted corresponding to adjacent tunneling to a piled raft. Through this, the change rate of settlement along the pile-length prediction model was developed. Then, to determine the reliability of the predictions, the $S_R$ prediction was compared and analyzed with the laboratory model test results, since $S_P$ could not be measured in the laboratory model test.

In the verification process of the models, the results of the laboratory model test were only inputted to prediction model_$S_R$ as training data, and the comparison of the predicted $S_R$ and actual values is shown in Figure 14. The raft-settlement prediction model in the laboratory model test using prediction model_$S_R$ showed an RMSE of 0.02 for the XGB algorithm; nevertheless, MLP was 8.23, which was significantly inaccurate. The RMSE of MLP was due to an imbalance in the number of samples, because those in the laboratory model test were 15, which is considered very small compared to the number of data used in the actual training.

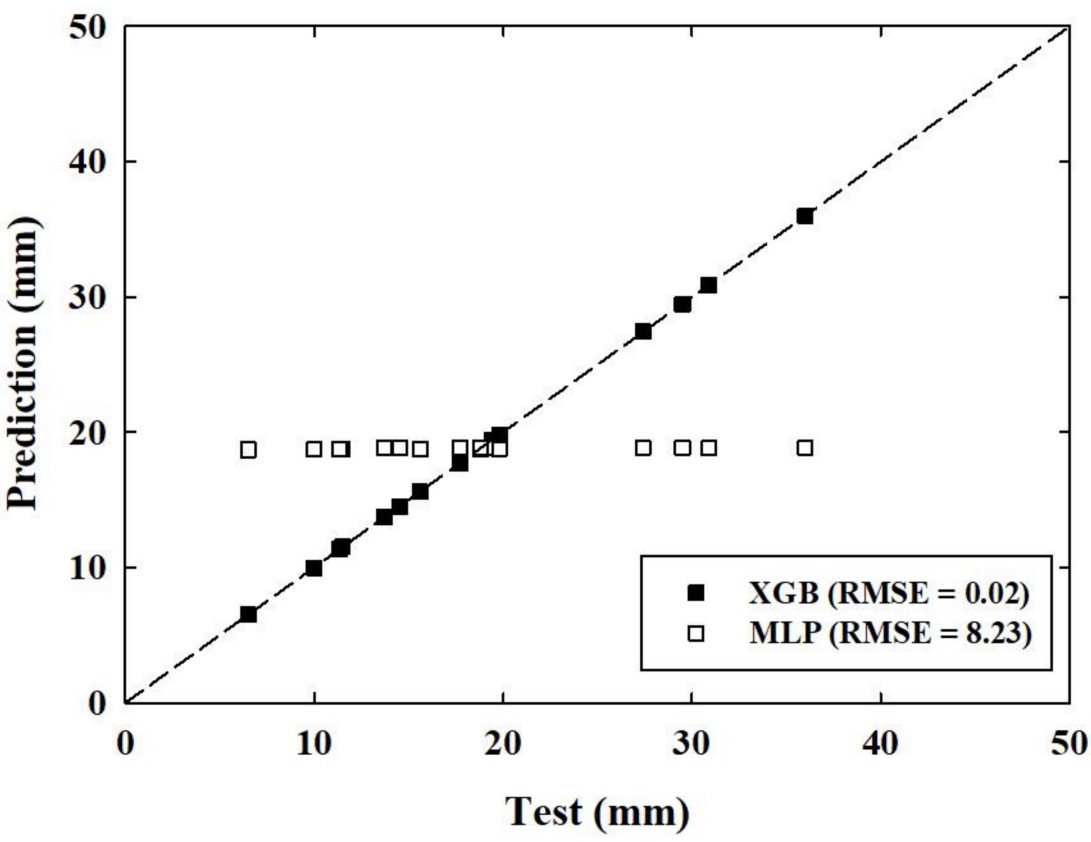

**Figure 14.** Verification of prediction model_$S_R$.

However, XGB had significantly less of an effect, which indicated that XGB is more reliable than MLP. The cause of this will be compared by obtaining more verification data.

### 5.3.3. Feature Importance

The utilized XGB in this study is a tree-based algorithm that has the advantage of quantitatively analyzing the effects of each feature in the preference model.

Figure 15 shows the normalization of feature importance derived through the XGB algorithm in each model. As shown in Figure 15a, $P_L$ had the greatest impact on prediction model_$S_R$ with 0.234, and $T_F$ and $O_V$ were 0.174 and 0.165, respectively. $O_H$, $E_G$, $P_{CTC}$, $B_P$, and $P_N$ were 0.118, 0.09, 0.074, and 0.071, respectively, and $B_r$ was 0.0. However, this should not be interpreted as $B_r$ having no effect on raft settlement due to tunneling.

$B_r$ is a variable that intuitively determines the adequacy of the model in the selection of variables, which is highly correlated to real $B_P$ and $P_L$. Therefore, the feature importance of $B_r$ being 0 should only indicate that this model is rational and does not mean that it does not affect raft settlement induced by tunneling.

The variables that affected preference model_$S_P$ are represented in Figure 15b.

$P_L$ resulted in 0.222, showing the highest influence in prediction model_$S_P$, and $T_F$ and $O_V$ were 0.152 and 0.148, respectively. $O_H$ was the next important feature in prediction model_$S_R$; however, $P_N$ was the next high F score in prediction model_$S_P$, with 0.106. $E_G$, $O_V$, $B_P$, and $P_{CTC}$ were 0.099, 0.097, 0.091, and 0.085, respectively.

The feature importance of prediction model_dS is shown in Figure 15c.

In this model, $O_H$ scored the highest with 0.183, followed by $T_F$, $O_V$, $P_L$, $E_G$, $P_{CTC}$, $B_P$, and $P_N$ with 0.170, 0.165, 0.150, 0.108, 0.093, 0.066, and 0.065, respectively. This shows that $T_F$ was the most common influential variable in all models. Thus, it is an important variable to consider when analyzing the settlement behavior of piled rafts due to tunneling.

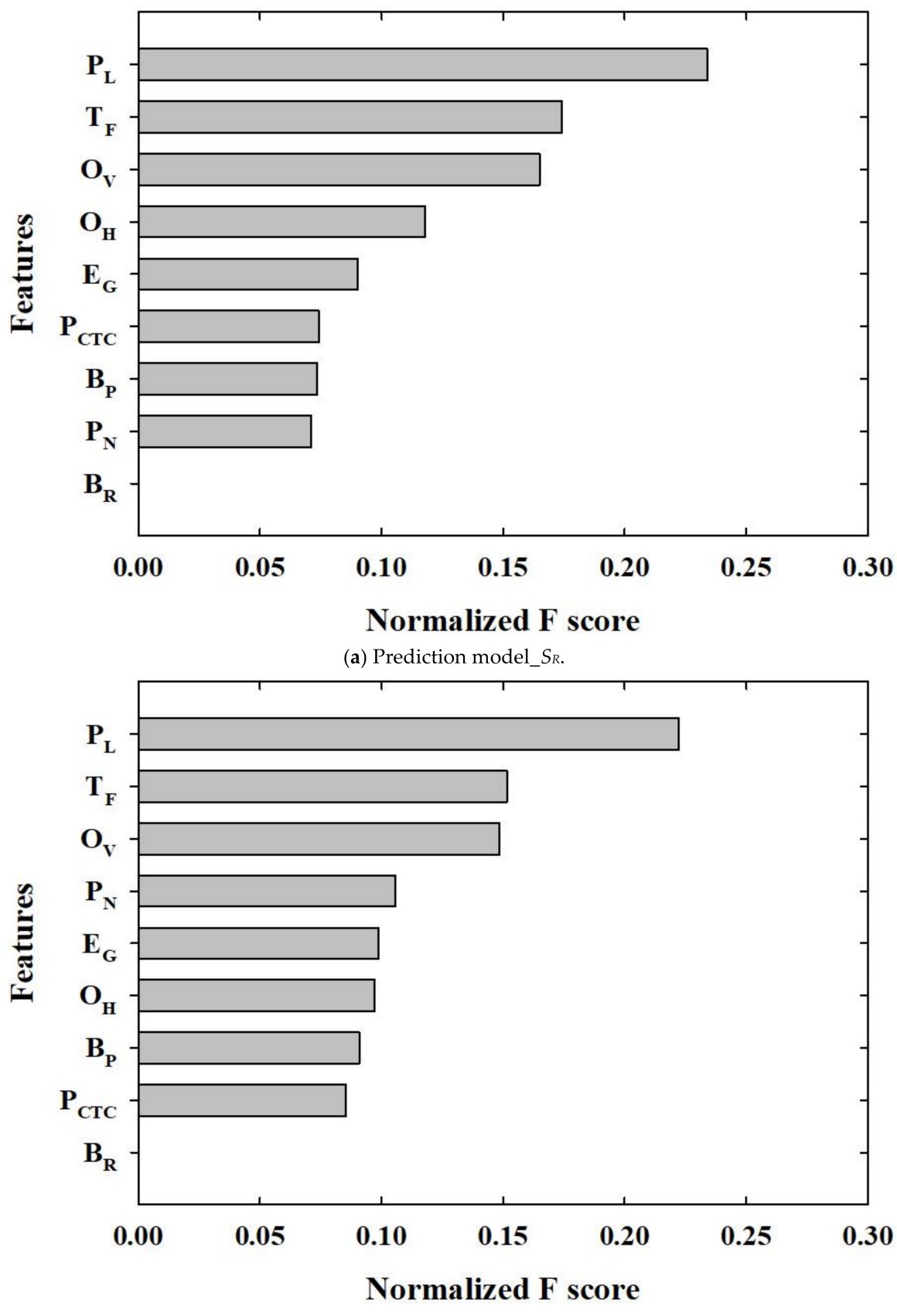

(**a**) Prediction model_$S_R$.

(**b**) Prediction model_$S_P$.

**Figure 15.** *Cont*.

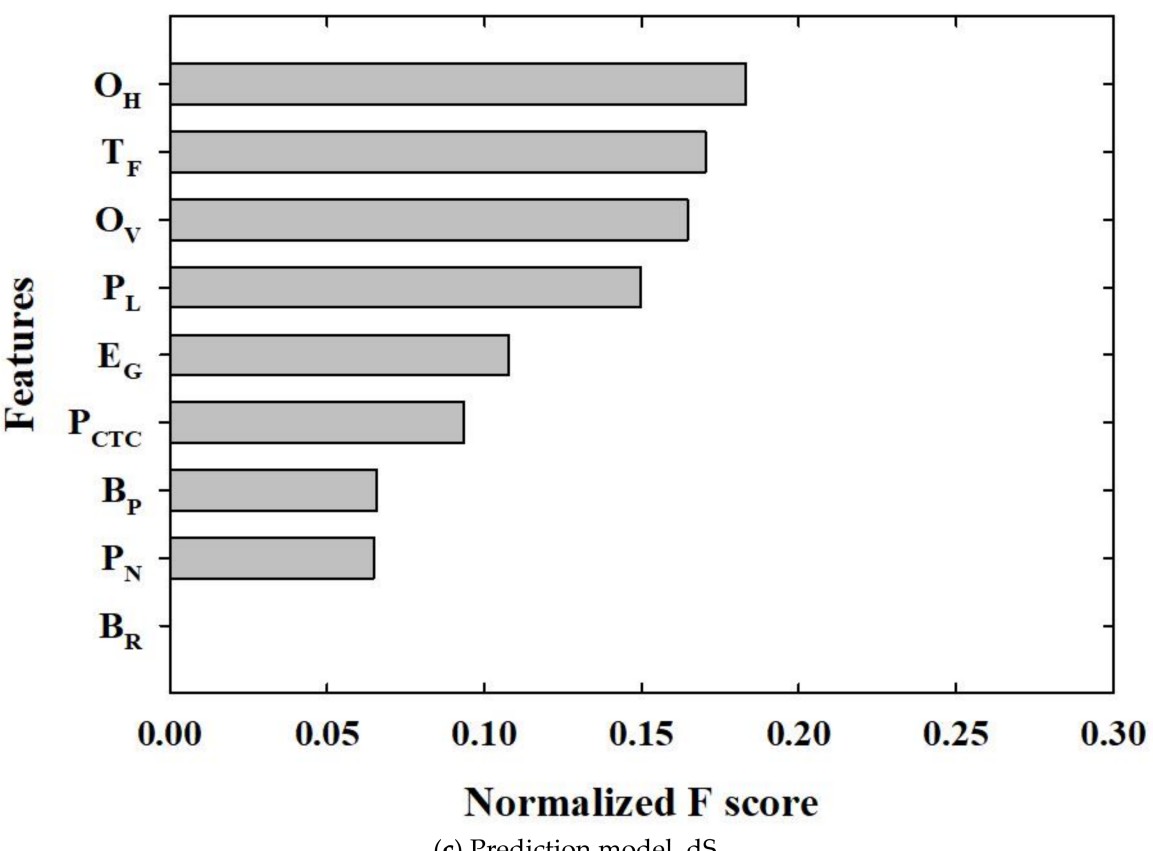

(**c**) Prediction model_dS.

**Figure 15.** Feature importance of prediction models.

## 6. Conclusions

In this study, the settlement behavior of piled rafts due to tunneling was analyzed, and prediction models were developed using machine learning. The prediction model was studied by dividing it into raft settlement and pile settlement. On this basis, the settlement change rate along the pile length was analyzed.

The raft settlement obtained through the laboratory model test was compared with the measured value using the developed prediction model for raft settlement. In addition, the influence of features on the prediction model of raft settlement, pile settlement, and change rate of settlement along the pile length was quantitatively analyzed through XGB.

- Laboratory model tests were performed on loose ground, and raft settlement was analyzed as the horizontal offset between pile toe and tunnel crowns increased, where the vertical offset between pile toe and tunnel crown was maintained at 0.5 D. $P_N$ being 1, raft settlement of 0.19, 0.16, and 0.07 mm occurred at 0.0, 1.0, and 2.0 D, respectively. When $P_N$ was 2, it was 0.31, 0.12, and 0.10 mm; at $P_N = 3$, it was 0.36, 0.20, and 0.11 mm. When horizontal offset between pile toe and tunnel crown was maintained at 0.0 D and vertical offset between pile toe and tunnel crown was 0.5, 1.0, 2.0 D, for $P_N = 1$, raft settlement of 0.19, 0.15, and 0.14 mm occurred; at $P_N$ 2, this was 0.31, 0.19, and 0.18 mm; and when $P_N$ was 3, this was 0.36, 0.30, and 0.27 mm, respectively.
- The prediction model of raft settlement showed better performance in XGB with 2.16 for XGB and 2.44 for MLP. As a result of applying this to the laboratory model test data, it showed a value of 0.02 for XGB, and 8.23 for MLP. This is considered to be an error caused by a significantly smaller number of laboratory model tests data counts compared to FE analysis data. This indicates that MLP is more reliant on the number of data compared to XGB.

In the prediction model of raft settlement derived through XGB, feature $P_L$ had an effect of 0.234, $T_F$ was 0.174, $O_V$ was 0.165, and $O_H$ was 0.118, showing an effect of more than 10%. On the other hand, $E_G$, $P_{CTC}$, $B_P$, and $P_N$ were 0.090, 0.074, 0.074, and 0.071, respectively, showing less than 10% effect.

- XGB and MLP had an RMSE of 1.63 and 2.32 in the prediction model of pile settlement, respectively. $P_L$ had the highest importance of 0.22, and $T_F$, $O_V$, and $O_N$ were 0.152, 0.148, and 0.106, respectively. These features affected the prediction model of pile settlement by more than 10%. $E_G$, $O_H$, $B_P$ and $P_{CTC}$, which were 0.099, 0.097, 0.091, and 0.085, respectively, affected this model by less than 10%.

- The prediction model of settlement change rate along the pile length is a concept in which the difference between raft and pile settlement due to tunneling is normalized by pile length, and it was assumed to be linear in this study. In the prediction model of settlement change rate along the pile length, the RMSE from the XGB algorithm was 0.05, and MLP was 0.07, respectively. In the obtained values from the prediction model, the maximum was 0.64 and the minimum was −0.71.

In the prediction model of change rate of settlement along the pile length, $O_H$ showed importance of 0.183, and $T_F$, $O_V$, $P_L$, and $E_G$ showed importance of 0.171, 0.165, 0.155, and 0.108, respectively. In the case of $P_{CTC}$, $B_P$, and $P_N$, importance of 0.094, 0.066, and 0.065 was less than 10%.

## 7. Discussion

The change rate of settlement could be predicted at the top and bottom of the pile for a piled raft induced by adjacent tunneling. In the case in which settlement at the bottom of pile was greater than the top, tension force occurred in the pile. This means that this affected the stability of the pile and upper structure because the pile consisted of concrete, which has very weak resistance against tension; therefore, the change rate of settlement along the pile length can be an important criterion for underground development in urban areas. The relationship between the change rate of settlement and pile axial force needs to be further investigated.

Machine learning is widely utilized for analysis and prediction in many academic and industrial categories and has a powerful application. In this study, the data from numerical analysis and laboratory model test are utilized. However, monitoring data from real construction sites will improve the model performance and be more meaningful.

The authors have planned to obtain settlement and axial force data from a real tunnel construction site, which will then be utilized as input data for the prediction model in this study. It is expected to improve and advance the usage of the model.

**Author Contributions:** Conceptualization, D.-W.O.; methodology, D.-W.O. and H.-J.P.; software, D.-W.O. and Y.-J.L.; validation, D.-W.O., H.-J.P., and S.-M.K.; formal analysis, D.-W.O. and S.-M.K.; investigation, D.-W.O.; resources, D.-W.O., S.-M.K., and Y.-J.L.; data curation, D.-W.O. and S.-M.K.; writing—original-draft preparation, D.-W.O.; writing—review and editing, H.-J.P.; visualization, S.-M.K.; supervision, Y.-J.L.; project administration, Y.-J.L. and H.-J.P.; funding acquisition, Y.-J.L. and H.-J.P. All authors have read and agreed to the published version of the manuscript.

**Funding:** This study was financially supported by the National Research Foundation of Korea (NRF) grant funded by the Korea government (Ministry of Science and ICT) under Grant number 2021R1A2C2013162.

**Institutional Review Board Statement:** Not applicable.

**Informed Consent Statement:** Not applicable.

**Data Availability Statement:** Publicly available datasets were analyzed in this study. These data can be found by following this link: https://drive.google.com/file/d/1qiXGaxRU4TUhhZkrfBT_j6 gGlkSyR44p/view?usp=sharing, accessed on 28 June 2021.

**Acknowledgments:** This study was financially supported by the Seoul National University of Science and Technology.

**Conflicts of Interest:** The authors declare no conflict of interest.

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
