# Peer review of "Prediction of Change Rate of Settlement for Piled Raft Due to Adjacent Tunneling Using Machine Learning"

_applsci, doi:10.3390/app11136009_

Round 1

Reviewer 1 Report

The article deals with current issues related to the probabilistic approach in geotechnical design through the use of advanced tools for predicting the behavior of soil-structures interaction. The publication is kind of mathematical modelling analysis based on results from the one small model (on a small scale). Of course, this is a good approach at the beginning, but it requires further analysis if it is to be useful for an application in practice. 

Notes and suggestions:

References to literature should be developed, indicating in more detail what they are about. The record like this: [1-10] is too general and not professional (from scientific point of view).

There is no comment on the merits of using the indicated constitutive model. Soil mechanics are currently based on descriptions of nonlinear soil behavior. The results are missing a comment: Does this simplified model affect the results?

With such sublime analysis on the scale model, it is expected to provide ground parameters from the own studies and not from the literature. The ground for analysis should be tested, some of the values adopted are debatable (e.g unit weight).

Add a broader description of tunnel modeling and hole details about presented experiment.

Technical note:

Figure 6 is not necessary in my opinion – it is enough to provide information in the text. 

Figure 7 is not clear enough, fonts are too small, it must be improved.

Author Response

Paper ID: applsci-1249405

Title: Prediction of Change Rate of Settlement for Piled Raft Due to Adjacent Tunnelling Using Machine Learning

The authors deeply appreciate all the invaluable comments from the reviewers. These comments will be carefully considered in the manuscript.

Comments from Reviewer 1:

The article deals with current issues related to the probabilistic approach in geotechnical design through the use of advanced tools for prediction the behavior of soil-structures interaction. The publication is kind of mathematical modelling analysis based on results from the one small model (on a small scale). Of course, this is a good approach at the beginning, but it requires further analysis if it is to be useful for an application in practice.

  1. References to literature should be developed, indicating in more detail what they are about. The record like this: [1-10] is too general and not professional (from scientific point of view).

This should be revised manuscript as the reviewer’s comment.

  1. There is no comment on the merits of using the indicated constitutive model. Soil mechanics are currently based on descriptions of nonlinear soil behavior. The results are missing a comment: Does this simplified model affect the results?

This should be added to manuscript as the reviewer’s comment, the reason the MC model is only used for this study.

  1. With such sublime analysis on the scale model, it is expected to provide ground parameters from the own studies and nor from the literature. The ground for analysis should be tested, some of the values adopted are debatable (e.g unit weight).

This should be revised manuscript as the reviewer’s comment.

  1. Add a broader description of tunnel modeling and hole details about presented experiment.

This should be added to manuscript as the reviewer’s comment, the reason the volume loss is only used for this study, not for tunnelling methods such as NATM, Shield etc.

  1. Figure 6 is not necessary in my opinion – it is enough to provide information in the text.

This should be revised manuscript as the reviewer’s comment.

  1. Figure 7 is not clear enough, fonts are too small, it must be improved.

This should be revised manuscript as the reviewer’s comment.

Reviewer 2 Report

This study used machine learning to predict the change rate of settlement for the piled raft. This is interesting research and I enjoy reading it. However, I do have a few comments about the quality of the presentation of this study. For example, figures 8 and 9 can be adjusted into a single page that would be easier to read and make a comparison.  Other figures can be arranged accordingly. 

The introduction is too short and it lacked the objective of this study, contribution, and how is this different from closely related other studies.  

The discussion section could be a separate topic that highlights the importance of the current study and compare it with the previous finding along with policy implications. 

Please make to make all tables and figures stand alone and are in the same format. 

Author Response

Paper ID: applsci-1249405

Title: Prediction of Change Rate of Settlement for Piled Raft Due to Adjacent Tunnelling Using Machine Learning

The authors deeply appreciate all the invaluable comments from the reviewers. These comments will be carefully considered in the manuscript.

Comments from Reviewer 2:

This study used machine learning to predict the change rate of settlement for the piled raft. This is interesting research and I enjoy reading it. However, I do have a few comments about the quality of the presentation of this study.

  1. For examples, figures 8 and 9 can be adjusted into a single page that would be easier to read and make a comparison. Other figures can be arranged accordingly.

This should be revised manuscript as the reviewer’s comment.

  1. The introduction is too short and it lacked the objective of this study, contribution, and how is this different from closely related other studies.

This should be revised manuscript as the reviewer’s comment.

  1. The discussion section could be a separate topic that highlights the importance of the current study and compare it with the previous finding along with policy implications.

This should be revised manuscript as the reviewer’s comment.

  1. Please make to make all tables and figures stand alone and are the same format.

This should be revised manuscript as reviewer’s comment.

Reviewer 3 Report

In this study, Oh et al. implement a machine learning model to predict the Change Rate of Settlement for Piled Raft Due to Adjacent Tunnelling. XGBoost and MLP have been assessed to consider which model worked well on this specific problem. Although the idea is of interest, there are some major points that need to be addressed:

1. Overall, English writing and style should be improved significantly. There are grammatical errors, typos, as well as jargon.

2. How did the authors tune the optimal hyperparameters of the models?

3. Some figures or tables should be merged together to contain significant information, and make the article more concise. For example, among Figs 7 to 14, Table 6, Table 7, ...

4. There are a lot of machine learning models or ensemble models, why did the authors only try with XGB and MLP?

5. Is there any baseline comparison among models?

6. The authors should compare the predictive performance with previously published studies on the same problem/data.

7. More discussions should be added.

8. Machine learning (i.e., XGB, MLP, ...) has been used in previous studies such as PMID: 32942564 and PMID: 33036150. Therefore, the authors are suggested to refer to more works in this description.

9. Quality of figures should be improved.

10. Why there are watermarks in the figures?

11. Source codes should be provided for replicating the methods.

Author Response

Paper ID: applsci-1249405

Title: Prediction of Change Rate of Settlement for Piled Raft Due to Adjacent Tunnelling Using Machine Learning

The authors deeply appreciate all the invaluable comments from the reviewers. These comments will be carefully considered in the manuscript.

Comments from Reviewer 3:

In this study, Oh et al. implement a machine learning model to predict the Change Rate of Settlement for Piled Raft Due to Adjacent Tunnelling. XGBoost and MLP have been assessed to consider which model worked well on this specific problem. Although the idea is of interest, there are some major points that need to be addressed:

  1. Overall, English writing and style should be improved significantly. There are grammatical errors, typos, as well as jargon.

This paper received the English correction service from MDPI. But this should be revised manuscript as reviewer’s comment.

  1. How did the authors tune the optimal hyperparameters of the models?

The GridSearchCV was used for optimization of hyperparameters in this study. But GridSearchCV is not special method for optimization of hyperparameters, so, the authors did not mention in the manuscript.

  1. Some figures or tables should be merged together to contain significant information and make the article more concise. For example, among Figs 7 to 14, Table 6, Table 7, ...

This should be revised manuscript as the reviewer’s comment.

  1. There are a lot of machine learning models or ensemble models, why did the authors only try with XGB and MLP?

The XGB is known as one of the most powerful algorithms that is based on tree. Particularly, the XGB is capable to quantitatively analyse the importance of each feature with F score. This indicates that, the algorithm has very powerful advantage for academic category (usage).

MLP is the algorithm which is widely utilized in artificial intelligence that is based on neural network. Moreover, many literatures paper in geotechnical engineering with AI, have utilized neural network algorithms for regression problem including Support Vector Machine (SVM), Multi-Layered Perceptron (MLP) etc. however, the number of researches conducted using XGB algorithms are much less than MLP., additionally, other algorithms based on neural networks including CNN, SVM, YOLO will be used and compared in further research using image contours analysis, and also settlement of raft and pile for piled raft due to tunnelling from field monitored data will be utilized for more advance and developed model.

  1. Is there any baseline comparison among models?

If reviewer ask vanilla code of each algorithm for general case such as Iris classification, California or Boston real state, they are searched easily on the website. In case of settlement of raft and pile for piled raft due to tunnelling problem is very specify, so there no vanilla code.

This study is first step and fundamental study for understanding the relationship about piled raft-tunnelling, the change of settlement for raft and pile tip. The results from this study can be baseline. But it’s not sure because only numerical analysis and model test was conducted and data from them were used for prediction model. If data from field monitoring is applied to the train data set, further advanced prediction model would be produced, however, the performance will be worse for sure. This is the reason we cannot define how much baseline is.

  1. The authors should compare the predictive performance with previously published studies on the same problem/data.

With all due respect, as the author already mentioned, research with respect to settlement of raft and pile for piled raft foundation due to tunneling using machine learning is the first. Of course, research about interaction behavior of single pile or grouped pile or piled raft foundation with tunneling have been conducted by so many researchers and engineers with considering a couple of features only. It is difficult to final studies that considered diverse variables that affect on settlement of piled raft including ground conditions, pile and raft elements, tunnel condition etc. with utilization of machine learning. This is the reason why prediction model of raft settlement are compared to result from laboratory model test for verification.

  1. More discussions should be added.

This should be revised manuscript as the reviewer’s comment.

  1. Machine learning (i.e., XGB, MLP,...) has been used in previous studies such as PMID: 32942564 and PMID: 33036150. Therefore, the authors are suggested to refer to more works in this description.

This should be revised manuscript as the reviewer’s comment.

  1. Quality of figures should be improved.

This should be revised manuscript as the reviewer’s comment.

  1. Why there are watermarks in the figures?

The commercial finite element method program used for this study is PLAXIS academic version made by Bentley Systems. Before the submission of this paper, the authors had asked the method to hide or delete watermarks, because those showed up suddenly after a new update in this year earlier. They responded that they decided to keep watermarks for PLAXIS academic version based on their policy.

  1. Source codes should be provided for replicating the methods.

With all due respect, the authors have planned to progress the further research be mentioned in the manuscript. If source codes are opened, hyperparameter of algorithms used in this study will be known to another researchers, it will be hard to purse the further research and our organization offered fund for this research.

Round 2

Reviewer 2 Report

It appears that the authors genuinely addressed all my comments and improved the manuscript considerably. I would ask the authors to move the conclusion section after discussion during the proofread.  This manuscript warrants publication in the applied science. Goodluck. Thanks

Reviewer 3 Report

My previous comments have been addressed well.